# The Biochemical Mechanism of Fork Regression in Prokaryotes and Eukaryotes—A Single Molecule Comparison

**DOI:** 10.3390/ijms23158613

**Published:** 2022-08-03

**Authors:** Piero R. Bianco

**Affiliations:** Department of Pharmaceutical Sciences, College of Pharmacy, University of Nebraska Medical Center, Omaha, NE 68198-6025, USA; pbianco@unmc.edu; Tel.: +1-402-559-8135

**Keywords:** RecG, SMARCAL1, HARP, SSB protein, RPA, stalled replication fork, fork regression, fork reversal, DNA repair, DNA replication

## Abstract

The rescue of stalled DNA replication forks is essential for cell viability. Impeded but still intact forks can be rescued by atypical DNA helicases in a reaction known as fork regression. This reaction has been studied at the single-molecule level using the *Escherichia coli* DNA helicase RecG and, separately, using the eukaryotic SMARCAL1 enzyme. Both nanomachines possess the necessary activities to regress forks: they simultaneously couple DNA unwinding to duplex rewinding and the displacement of bound proteins. Furthermore, they can regress a fork into a Holliday junction structure, the central intermediate of many fork regression models. However, there are key differences between these two enzymes. RecG is monomeric and unidirectional, catalyzing an efficient and processive fork regression reaction and, in the process, generating a significant amount of force that is used to displace the tightly-bound *E. coli* SSB protein. In contrast, the inefficient SMARCAL1 is not unidirectional, displays limited processivity, and likely uses fork rewinding to facilitate RPA displacement. Like many other eukaryotic enzymes, SMARCAL1 may require additional factors and/or post-translational modifications to enhance its catalytic activity, whereas RecG can drive fork regression on its own.

## 1. Introduction

The duplication of the genome is inherently accurate and highly processive [1,2]. In *E. coli*, DNA replication is initiated at *ori*C, generating two replication forks that move bidirectionally away from one another until their progress is permanently impeded in the terminus region, with the two daughter molecules subsequently resolved via site-specific-recombination or decatenation [3,4,5,6,7]. In contrast, in eukaryotic cells, DNA replication is initiated at multiple origins on each linear chromosome with the progression of forks continuing until another fork, coming in the opposite direction, is encountered, at which point termination occurs and the replisome is disassembled [8,9,10,11].

However, DNA replication forks seldom make it to their endpoint without encountering problems that could be potentially life-threatening to the cell [12,13,14]. In *E. coli*, each fork is thought to stall or collapse entirely at least once per cell cycle and possibly even more frequently [13,15,16,17,18,19,20,21,22,23]. Fork stalling is the result of the advancing replisomes encountering physical impediments to progression or the replisomes experiencing a shortage of DNA synthesis precursors [24,25,26]. Impediments to progression include proteins bound to the DNA ahead of the replication fork, such as repair enzymes, repressors, or RNA polymerase (either alone or coupled to ribosomes); non-coding lesions in the template DNA; unusual secondary structures that arise in the DNA; R-loops; and either single- or double-strand breaks [21,22,23,24,25,27,28,29]. In eukaryotes, fork stalling occurs in response to, at a minimum, DNA alkylation damage, nucleotide depletion, as well as to polymerase inhibitors and programmed replication fork barriers [8,24,30,31,32]. Impeded forks must be rescued and a failure to accomplish this is a lethal event. Consequently, the accurate and faithful duplication of the genome relies on the DNA replication, repair, and genetic recombination machinery working closely together [14,15,21,33,34,35,36].

An impeded fork is one in which net forward progression has been prevented (Figure 1A). Generally, there are two kinds of impeded forks. In either of these impeded states, polymerase uncoupling and/or the dissociation of one or more replisome components can occur [37]. This requires the reloading of dissociated components once the DNA damage has been addressed so that DNA replication can resume. The first kind of impeded fork is termed collapsed or broken, and this can arise from single- or double-strand breaks (Figure 1B,C). Here, one or more fork arms are no longer connected to the parental duplex, and the resurrection of the fork architecture requires the activity of multiple enzymes [14,36,38]. As this type of impeded fork cannot be rescued by regression, it will no longer be discussed herein.

The second type of impeded fork is defined as one whose forward progress has stalled. Here, the fork architecture remains intact, may contain single-stranded regions in either one or both arms, and the progression of the replisome is blocked. In this situation, fork rescue involves moving the fork in a net backward direction in a reaction away from the impediment, in the direction opposite to that of replisome movement, while simultaneously producing DNA structure(s) which, upon further processing, result in the reloading of the replisome. This reaction is known as fork regression (Figure 1D). Once the DNA damage has occurred, the fork position can be reversed back to the approximate original position and/or structure, in a reaction known as fork reversal.

For fork regression to occur, a specialized DNA helicase is required. First, this enzyme must couple the unwinding of the nascent leading and lagging strand arms of the fork to duplex rewinding (Figure 1E). This rewinding regenerates the parental duplex in the wake of the enzyme and, if the nascent arms contain sufficient regions of complementary dsDNA, an additional region of duplex rewinding will be observed that is extruded ahead of the advancing enzyme. The coupled unwinding/rewinding of DNA converts the 3-arm fork into a 4-arm chicken foot intermediate or Holliday junction. The Holliday junction structure is the central intermediate in most fork rescue models [39]. Second, for efficient regression to occur, the enzyme must bind to the fork with a stoichiometry of one (either a monomer or single oligomeric complex), and furthermore, it must by necessity bind in only one orientation so that unidirectional regression is facilitated. Finally, the enzyme must generate sufficient force during translocation, DNA unwinding, and duplex annealing, so that any protein obstacle is readily displaced (Figure 1F). These obstacles could be single-stranded DNA binding proteins or duplex DNA binding proteins such as repressors, nucleosomes, or nucleoid-associated proteins loading back onto the DNA at the incorrect time (before forks have been completed).

In *E. coli*, the enzyme responsible for fork regression is the DNA helicase known as RecG, which has been well-characterized in vivo, in vitro, and at the single-molecule level [40,41,42]. Although there is no known homolog of RecG in mammalian cells, there is a nuclear-encoded ortholog in plants [43]. RECG1, however, functions in mitochondrial and plastid genome stability only. In eukaryotic cells, as many as nine enzymes have been proposed to drive fork regression [36,44]. However, only one of these, SWI/SNF-related matrix-associated actin-dependent regulator of chromatin subfamily A-like protein 1 (SMARCAL1; also known as HARP), has been studied in bulk-phase and at the single-molecule level and therefore will be compared to RecG [45,46,47]. There is no known homolog of SMARCAL1 in prokaryotes.

## 2. Fork Regression Enzymes

### 2.1. RecG

The RecG protein was identified as a mutation that mildly affected recombination and survival following UV irradiation [48]. Subsequent studies demonstrated that it participates in all three pathways of recombination (RecBCD, RecE, and RecF) and that it has an overlapping function with the products of the *ruvA* and *ruvB* genes [49,50]. The purified enzyme possesses ATPase and DNA helicase activities [49,51]. It has been classified as a member of the SF2 DNA helicases and nucleic acid translocases [52].

In vitro, RecG unwinds DNA in a 3′ → 5′ direction [53,54]. It is active as a monomer on forks with either single-stranded or duplex arms, as well as Holliday junctions [55]. Several bulk-phase studies have shown that this helicase efficiently converts stalled replication fork substrates into structures that can be acted upon by additional members of the recombination machinery [41,56,57,58,59,60]. Despite its ability to process fork substrates, its role in fork regression was hotly contested with that of RuvAB, the bacterial branch migration complex, which has been proposed to process forks instead [61,62]. However, careful biochemical analyses demonstrated that RecG acts at stalled replication forks first, outcompeting a vast excess of RuvAB, and can displace the single-strand DNA binding protein in the process [55,63,64].

Further support for the role of RecG in fork regression came from the Wigley group, who produced a clear crystal structure of the enzyme bound to a model fork [60]. Their study provided both additional mechanistic insights into how RecG processes a fork, as well as indicated how other enzymes may function as well. The structure showed that the enzyme is divided into two general domains, connected by a long α-helical linker. Domain I comprises the N-terminal half of the protein and contains the wedge domain, which includes the oligonucleotide-oligosaccharide binding fold (OB-fold; Figure 2). The wedge domain is essential for specific binding to branched DNA structures and it is also intrinsic to DNA strand separation [65]. In addition, the OB-fold plays a central role in helicase regulation by the SSB protein [63,66,67,68,69]. Domain II consists of the helicase motifs, which couple the energy associated with ATP binding, hydrolysis, and product release to enzyme motion, DNA unwinding and rewinding, and fork clearing [60,70,71].

Although RecG binds to and processes forks efficiently in the absence of SSB protein, SSB enhances the loading of the helicase onto forks, resulting in remodeling of the helicase in the process [67]. Here, the binding of the PXXP motifs in the linker domain of SSB to the OB-fold of RecG is essential in the loading of the helicase onto DNA [69]. These interactions are also key to remodeling as binding precludes the interaction of the wedge domain with the fork, so the interaction of SSB-remodeled RecG with nucleic acid is mediated by the helicase domains of the enzyme to the parental duplex ahead of the fork. In addition to the loading of the enzyme onto forks, SSB also stabilizes RecG on ssDNA and forks [63,64]. Finally, although SSB binds to ssDNA with high affinity, RecG efficiently displaces the protein from DNA and this displacement requires the linker domain of SSB to bind to adjacent SSB OB-folds and not to that of RecG [73].

### 2.2. SMARCAL1

SWI/SNF-related, matrix-associated, actin-dependent regulator of chromatin and subfamily A-like 1 protein (SMARCAL1) is a member of the distant group of ATP-dependent chromatin remodeling factors, one of the six phylogenetic groups of remodeling enzymes [74]. These enzymes have been classified as DNA helicases due to the presence of the conserved helicase motifs [75]. However, similarl to many proteins that contain these motifs, chromatin remodelers do not exhibit DNA helicase or strand separation activity. Instead, they utilize the energy from ATP binding, hydrolysis, and product release to alter the architecture of chromatin [76].

SMARCAL1 was purified as a 68 kDa protein from a calf thymus and classified as DNA-dependent ATPase A [77]. Later, improved purification techniques resulted in the isolation of the full-length 105 kDa protein [78]. The human homolog was subsequently purified from HeLa cells and was initially called Hep-A-related protein or HARP due to its sequence similarity to the prokaryotic HepA protein [79]. Subsequent sequence analysis revealed homology to the ATP-dependent chromatin remodelers, resulting in the redesignation of the enzyme as SMARCAL1 [80]. In vitro studies have gone on to show that the ATPase activity is also stimulated by fork DNA structures and is an ATP-dependent annealing enzyme, although the authors called SMARCAL1 an ATP-dependent annealing helicase, which is not only confusing but misleading, as it cannot strand-separate, i.e., it has no helicase activity [81,82,83]. Even more confusing is the classification of SMARCAL1 as a chromatin remodeler, as to date this activity has not been associated with the protein. Instead, it is also thought of as a transcription co-regulator [80].

However, this multifunctional protein also plays a role in genome stability, as bulk-phase studies have shown that it can catalyze fork regression, generating Holliday-junction-like structures [47]. Although these studies are consistent with its role in maintaining genome stability in vivo [84,85,86], there are many enzymes capable of catalyzing fork remodeling using oligonucleotide-length substrates in vitro (for a review, see [36]). Furthermore, SMARCAL1 binds to the 32 kDa subunit of replication protein A (RPA), and this interaction is required for localization to forks in vivo [84,87,88]. Although RPA stimulates SMARCAL1 fork reversal activity when it is bound to an ssDNA gap on the leading template strand, it inhibits SMARCAL1 when bound to a replication fork with an ssDNA gap on the lagging strand [46]. This difference may be related to the polarity of fork binding by RPA, as this protein prefers forks with 3′-tails [89,90].

Thus, like RecG, SMARCAL1 shares many attributes that indicate that this enzyme has a role in fork regression. It can drive fork regression activity in vitro; it binds to its cognate single-strand DNA binding protein, which regulates its activity; and its substrate preference is for a fork with a gap in the leading strand. However, important differences exist. SMARCAL1 does not bind forks with a stoichiometry of 1 (whereas RecG does), and if it does oligomerize, this form is unknown. Instead, stoichiometry cannot be achieved in vitro, indicating either oligomerization or promiscuous DNA binding activity. Furthermore, and in contrast to RecG, it does not bind to a fork specifically in one orientation, as shown by footprinting. Instead, SMARCAL1 binds, in separate reactions, in two orientations to both the parental duplex and nascent leading/lagging strand arms. It is conceivable that the presence of the OB-fold in RecG enables both fork-specific and orientation-dependent binding by this enzyme, whereas SMARCAL1 does not contain a defined OB-fold. Finally, the wild-type SSB protein does not inhibit RecG, whereas RPA inhibits SMARCAL1 in a substrate-dependent manner (as explained in Section 5).

## 3. Method to Study Fork Regression at the Single-Molecule Level

To study fork transactions at the single-molecule level, a unique approach is required. This is because the reaction is isoenergetic, with equal numbers of base pairs being unwound as there are being rewound. To address this issue, the Croquette group developed an elegant hairpin approach and combined this with magnetic tweezers, a single-molecule technique they had pioneered several years earlier [91,92]. This combined method was then used to study two fork-specific enzymes, *E. coli* RecG and bacteriophage T4 UvsW [41].

The experimental system is shown in Figure 3A. The substrate used was a 1200 bp hairpin that was attached to the coverslip surface via multiple digoxygenin (dIG)/anti-dIG interactions. This makes sense because a single dIG/anti-dIG complex is disrupted in 0.015 s at 30 pN of force, and is therefore not sufficiently strong to withstand the forces associated with the experiments [93]. At the opposite end, the hairpin was attached to a superparamagnetic bead via a single avidin-biotin bond, which required 1700 s for disruption at the same force [94]. Once sandwiched between the bead and the coverslip, the hairpin was positioned on the stage of a magnetic tweezer instrument and a force field was applied. This magnetic field was used to control bead position and thus the Z-height above the coverslip surface. Simultaneously, it was used to prevent the hairpin from reannealing so that the reaction direction (regression vs reversal) could be controlled. To determine the bead position and thus the Z-height with single-base-pair resolution, an LED beam was shone onto the bead and the resulting image was projected onto the chip of a CCD camera (Figure 3A, images inset on the right, middle). At the same time, a reference bead was attached to the coverslip surface. As it was stationary, it acted as a measurement reference point for the mobile, DNA-attached bead. As the mobile bead changed position due to enzyme action, the image projected onto the camera changed, and, using software, precise measurements of position could be ascertained.

The approach using this substrate and magnetic tweezer setup is straightforward. If an enzyme binds to the hairpin in its partially unwound starting position and ATP is introduced, it can either regress or reverse the fork. Fork reversal occurs when the enzyme translocates from right to left on the substrate as shown, and this is visualized as the unwinding of the duplex arm of the hairpin (the parental duplex ahead of the work). As a result, the single-strand arms increase in length, and accordingly the Z-height increases as a function of time (Figure 3A, top right). In contrast, fork regression occurs when the enzyme translocates from left to right and this is visualized as a rewinding of the single-stranded DNA arms. As a result, the ssDNA arms decrease in length, and accordingly, the Z-height decreases as a function of time (Figure 3A, bottom right). Studies using this hairpin assess the ability of the enzyme to anneal two ssDNA arms (i.e., rewind the duplex) and also to determine the reaction direction (regression or reversal).

By annealing oligonucleotides in separate experiments, to either the upper or lower ssDNA arms, substrates with gaps in the lagging or leading strands can be created in situ, followed by the introduction of enzymes and ATP (Figure 3B). Studies using these forks can test whether an enzyme can couple dsDNA unwinding to fork rewinding, and also ascertain the fork preference of the enzyme. Finally, a more complex fork with three duplex arms can also be created in situ to evaluate the ability of an enzyme to regress a fork into a Holliday junction (Figure 3C).

## 4. Fork Arm Rewinding—An Essential Step in Fork Regression

Using the 1200 bp hairpin, the ability of RecG and separately SMARCAL1 to catalyze annealing was assessed [41,46]. At an opposing force of 17 pN, RecG catalyzed an efficient reaction, visualized as a decrease in Z-height as a function of time (Figure 4A). The multiple events observed are all from a single RecG molecule, catalyzing fork regression for 100–1200 bp per binding event. Analysis of multiple reactions revealed that RecG catalyzes reactions in the fork regression direction 99% ± 1% of the time (Figure 4B). Thus, this enzyme is exclusively a fork regression DNA helicase. During this reaction, RecG translates at a rate of 269 ± 2 bp/s and with a processivity of 480 ± 20 bp per binding event. As RecG utilizes 1 ATP to translocate 3 bp, 160 nucleoside triphosphate molecules are hydrolyzed on average per fork regression event [95]. Both the single-molecule and bulk-phase data demonstrate that the major sites of RecG binding and the production of translocation result from interactions with the parental duplex arm of the fork [41,95].

In contrast to the robust RecG, at 14 pN of opposing force, SMARCAL1 is a poor enzyme, capable of limited annealing events ranging from 1 to 4 footprint distances (i.e., 20–40 bp), which are repeated multiple times (Figure 4C,D). In some cases, SMARCAL1 disengages from the DNA, which then snaps back to the starting position (the reverse trajectory is 90° vertical, as in the far left of Figure 4D, and also the central trajectory). In other cases, such as those on the far right, the enzyme either cannot bind to the fork tightly enough to drive a unidirectional reaction and thus allows the reaction to slip back, or it switches fork arms and drives the reaction in the direction of reversal. Consequently, these data indicate that SMARCAL1 is a poor regression enzyme (compared to RecG) or, due to its ability to bind the fork at multiple sites possibly coupled to a low affinity for DNA, either disengages or switches sites and drives fork reversal as well [46].

## 5. Effects of Single-Stranded DNA Binding Proteins

A key aspect of the activity of a fork regression enzyme is the ability to work against large opposing forces so that protein obstacles do not impede the repair process. At *E.coli* replication forks, there is, on average, 0.5 to 1 kb of ssDNA available [4]. Using a site size of 40 nucleotides occluded per SSB tetramer, there would be, on average, 25 tetramers bound per fork, and these are bound tightly and cooperatively, predominantly to the lagging strand, with a K_d_ = 4 to 145 nM [96,97,98]. As SSB requires, at a minimum, 10 pN of force to displace a single tetramer, RecG must be able to work against at least this level of force [99]. Similarly, RPA will be bound to ssDNA present on the lagging strand with a minimum of 33 heterotrimers, bound using a site size of 30 nucleotides occluded, and with a K_d_ ranging from 15.29 ± 5.52 to 50 nM [100,101,102]. Even though RPA protects ssDNA by binding with high affinity, it is dynamic at DNA forks. As a consequence, fork rewinding induces RPA sliding on the ssDNA arms in front of the zippering fork [103]. Thus, there is finely balanced competition between the continuous association and dissociation of RPA at the fork and this is dictated by the fork itself and the applied forces. At low force (≤11 pN), the fork reanneals and RPA is displaced, whereas at higher forces the fork is unwound and RPA binding is favored.

To assess the effects of tightly bound ssDNA-binding proteins, it was first necessary to determine the force regime within which RecG could operate. Results have shown that RecG catalyzes duplex rewinding against forces of up to 35 pN, with only a moderate drop of about 40% in the rate (Figure 5A). This demonstrates that RecG is a very powerful monomeric motor that should be able to displace SSB. To test this, the hairpin was completely unzippered by the application of force, SSB was introduced and allowed to bind, resulting in the shortening of the ssDNA tether due to the wrapping of the polynucleotide around the SSB tetramer (Figure 5B) [104,105,106]. Even though SSB binds to ssDNA with high affinity, the wild-type protein poses little threat to the ability of RecG to regress the fork as the enzyme was capable of coupling rewinding to efficient SSB displacement at rates comparable to those that occur in the absence of SSB (Figure 5B). It is worth noting that following the introduction of RecG and ATP into the reaction chamber, there is a short lag before the onset of reannealing (Figure 5B,C). For the wild type, the lag is ~2 s and this increases as much as 14-fold when the linker region of SSB is either partially or completely deleted [107] (Figure 5C). As the binding of PXXP motifs from the linker of one SSB monomer is required for cooperative binding to the OB-fold of a monomer in an adjacent tetramer [69], these results provide an insight into the mechanism of SSB displacement by RecG. First, the binding of SSB to RecG is mediated by linker/OB-fold binding [69]. Second, these interactions are likely not directly involved in helicase displacement, as the RecG OB-fold is already bound to the fork. Third, this result reflects the ability of RecG to dislodge SSB from the DNA. When linker/OB-fold, SSB-SSB interactions are functional, as they are for the wild type and SSBΔC8 (an SSB mutant lacking the last eight residues), pushing by the translocating RecG is communicated between the tetramers, facilitating SSB displacement. In contrast, when SSB-SSB linker/OB-fold interactions are absent, as in SSB125 (the entire linker is deleted), these tetramers function as separate, tightly bound entities that impede RecG translocation.

For SMARCAL1 and RPA the situation is different. In the absence of the single-strand DNA binding protein, SMARCAL1 translocates approximately 20 bp or a footprint before dissociating (Figure 5D, bottom trace). In the presence of RPA, this distance increases to a SMARCAL1 footprint + RPA site size (20 + 30 nt), and similarly to reactions with SMARCAL1 alone, both regression and reversal reactions catalyzed by the same protein molecule are observed (Figure 5D, top trace). At 14 pN of opposing force, the enzyme rewinds DNA at rates of 200 bp/s, comparable to RecG but processivity is low (Figure 5E,F). At 3 pN of force, and in the presence of RPA, rewinding rate is reduced 2-fold but processivity is increased almost 10-fold. This is consistent with RPA stabilizing SMARCAL1 on the DNA and this may involve RPA-DNA binding, as well as SMARCAL1-RPA binding [87,88,108]. Even though SMARCAL1 and RPA bind one another, displacement does not require this interaction as RPA is displaced by SMARCAL1-Δ34, which lacks the RPA binding domain (Figure 5E,F). It is conceivable that fork zippering provides an assistive role in displacing RPA [102]. This is complex to dissect, as RPA heterotrimers may stabilize SMARCAL1 by limiting dissociating and by binding to other fork regions [102] and the heterotrimer at the fork is the first RPA to be displaced by fork zippering, coupled to SMARCAL1 translocation and “pushing” [87,88,103,108]. This first RPA may communicate with adjacent heterotrimers, facilitating displacement comparable to what is observed for RecG and SSB. More studies are required to dissect these dynamic interactions for both eukaryotic and prokaryotic systems.

## 6. Formation of the Central Intermediate of Fork Regression

Central to many models of fork rescue is the formation of an intermediate resembling a Holliday junction and sometimes referred to as a “chicken-foot” intermediate, which results from the unwinding and rewinding of an impeded fork (Figure 1D; [39]). The question of how this unwinding results in Holliday junction formation was unclear. Early studies of RecG by the Lloyd group demonstrated that RecG was capable of catalyzing this reaction [109]. Further work from this group and others showed that RuvAB does not drive this reaction [55,109,110,111]. Separately, work from the Cozzarelli laboratory showed that the accumulated positive torsional stress ahead of an advancing fork, when released at stalling, catalyzed Holliday junction formation [112]. Studies in eukaryotic cells showed a similar propensity and, in addition, several enzymes were proposed to catalyze this reaction [44,113].

To test whether a fork rescue DNA helicase could catalyze this reaction, the hairpin substrate was modified in situ to create a 600 bp, 3-duplex arm fork (Figure 3C). Following the introduction of RecG and ATP, RecG catalyzed an efficient, unidirectional, and processive regression reaction (Figure 6A). Once the helicase dissociated from the DNA, it branch-migrated back to its original fork structure induced by the 8 pN of force used to hold the substrate in place. Then RecG bound again and the reaction ensued. For RecG to catalyze Holliday junction formation, it coupled the unwinding of the forks arms to duplex rewinding in its wake and ahead of the translocating enzyme (Figure 1E). This result is significant, as it demonstrated the formation of an HJ or chicken foot intermediate by single RecG enzyme, a key intermediate in most fork rescue models [39,114]. For RecG, binding resulted in monotonic branch migration, terminating upon enzyme dissociation.

Using an identical substrate, the ability of SMARCAL1 to regress a fork into a Holliday junction was assessed. The results showed that, as for fork unwinding, this enzyme can catalyze fork regression, resulting in the formation of a Holliday junction (Figure 6B). This was the first demonstration of DNA helicase activity for SMARCAL1. However, the eukaryotic enzyme is poor in comparison to RecG, driving the reaction in the regression and then the reversal direction (in a time of 42–88 s), before finally executing a reaction with increased processivity. This is then followed by a lengthy reversal reaction where the Holliday junction is branch-migrated back to ~75% of the starting length. The inefficient reaction driven by SMARCAL1 (as compared to RecG) could be due to the promiscuous nature of the eukaryotic protein, a low affinity for DNA (that may be due to the absence of an identified OB-fold), and/or the absence of one or more regulator proteins or post-translational modifications. This is similar to what is observed in DNA strand exchange reactions when comparing *E. coli* RecA to the eukaryotic Rad51. RecA performs this reaction efficiently, requiring only SSB to remove the residual ssDNA secondary structure, whereas Rad51 requires both RPA and, at a minimum, the dsDNA translocase Rad54 [115,116].

## 7. Conclusions

The rescue of stalled DNA replication forks is essential to the viability of cells. The inability to accurately repair them could have disastrous consequences [12,13,14]. Here, the activity of two fork regression enzymes at the single-molecule level has been compared. Both enzymes possess the intrinsic characteristics required to drive fork regression and can displace protein obstacles, albeit using distinct mechanisms. In addition, both RecG and SMARCAL1 can couple DNA unwinding to duplex rewinding of a fork substrate to generate the central intermediate of many fork rescue models. It is conceivable that the discovery of fork regression mediators will enable SMARCAL1 to catalyze more efficient reactions and may regulate the reaction direction so that the enzyme is unidirectional. Alternatively, the bidirectional character of SMARCAL1 may be taken advantage of and different mediators may ensure fork processing in one direction or the other, as dictated by the stage of fork rescue. Finally, the hairpin-magnetic tweezer approach is amenable to the study of the many fork processing enzymes and may serve to further delineate the role that these proteins play in the different stages of fork rescue.

## Figures and Tables

**Figure 1 ijms-23-08613-f001:**
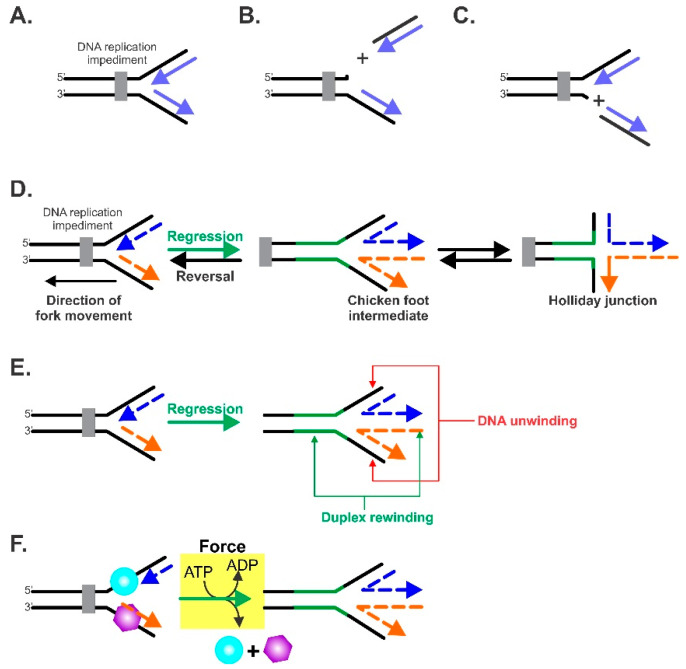
The key elements of the fork regression reaction. (**A**–**C**) An impeded fork is shown in (**A**) with the impediment, leading to cleavage of the nascent leading (**B**) or lagging strands (**C**). The net result is that the architecture of the fork is lost. (**D**) The terminology of the fork regression reaction. A DNA replication fork is shown impeded by an obstacle. To facilitate repair, the position of the fork is moved in the direction opposite to that of the fork’s movement, that is, in a net backward direction. This is defined as fork regression. The product of regression is known as the “chicken foot intermediate” which is essentially a Holliday junction. The opposite of fork regression is fork reversal, which takes place once the fork impediment has been removed. Reversal results in the restoration of the nascent fork structure, which ultimately results in the resumption of DNA replication. Black strands, parental DNA; blue and orange dashed strands indicate nascent leading strands and lagging strands, respectively; green strands represent nascent annealed parental strands. (**B**) The DNA enzymatic requirements of fork regression. For an enzyme to catalyze an efficient reaction, it must be an atypical DNA helicase that unwinds the nascent heteroduplex arms of the fork and couples this unwinding to duplex reannealing that occurs both in the wake of the enzyme, as well as ahead of the advancing molecular machine. (**C**) The force requirements of fork regression. The atypical DNA helicase must couple the energy associated with ATP binding, hydrolysis, and product release to the generation of mechanical force required to both regress the fork and displace proteins bound to single- (cyan sphere) or double-stranded (purple hexagon) regions of the fork. (**E**) To catalyze regression, the enzyme must couple DNA unwinding (red arrows) to duplex rewinding (green arrows). (**F**) In addition to being able to function as an atypical DNA helicase, the enzyme must also generate sufficient force to displace single- and/or double-strand DNA binding proteins during the fork regression reaction.

**Figure 2 ijms-23-08613-f002:**
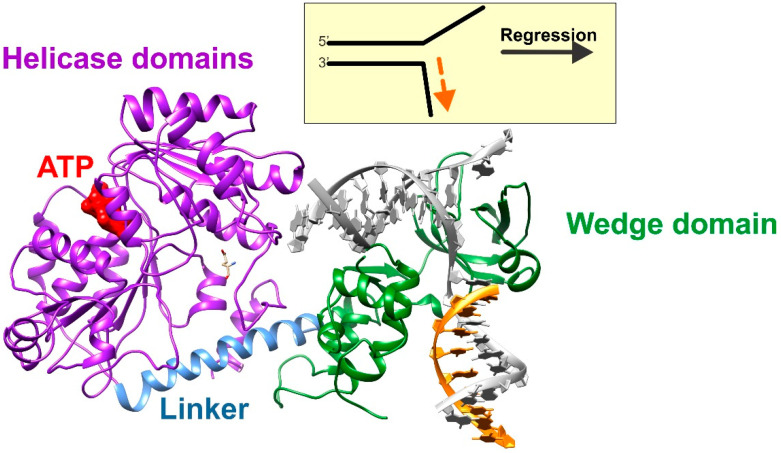
RecG binds to forks to catalyze regression. A ribbon diagram of a homology model of *E. coli* RecG bound to a fork substrate. The relevant regions are colored for clarity and the DNA strands are colored as in Figure 1. For comparison, a schematic of the fork is shown above, with DNA strand coloring the same as that in the structure. The model was built using Swiss-Model and PDB file 1GM5 as a template [60,72].

**Figure 3 ijms-23-08613-f003:**
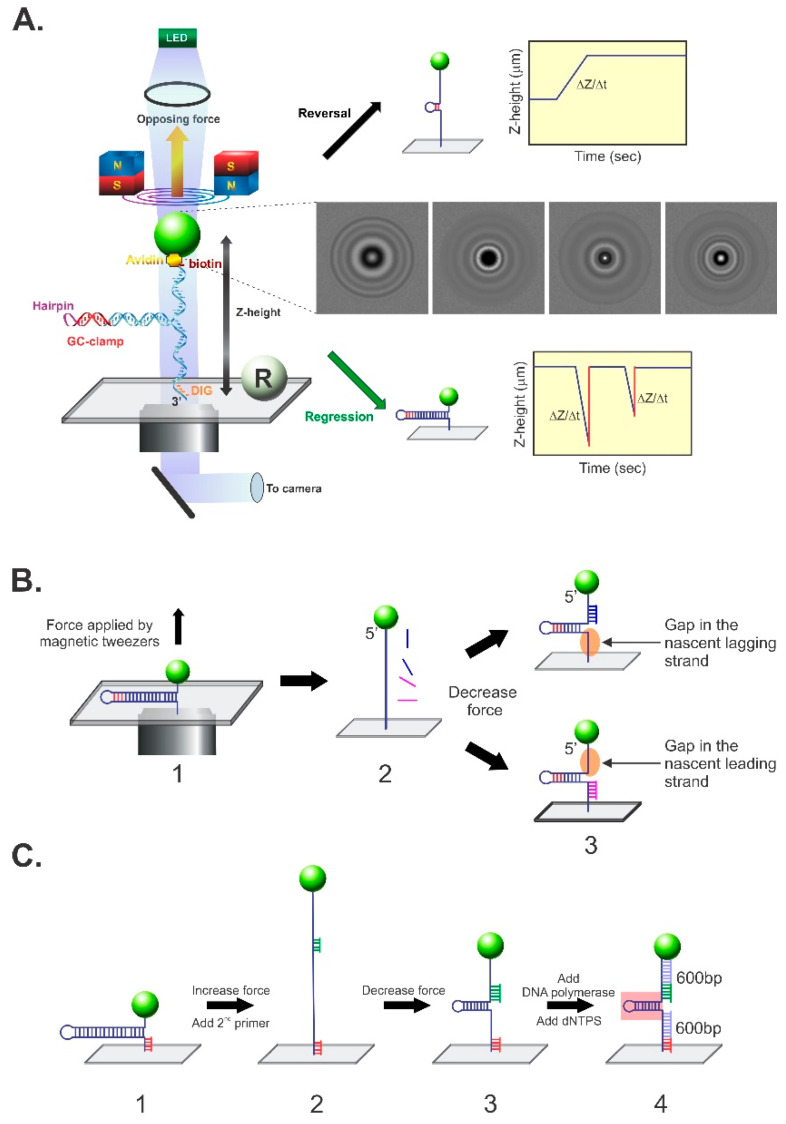
The magnetic tweezer, single-molecule assay that is used to dissect fork transactions. (**A**) Schematic of the assay developed by the Croquette group to monitor changes in the DNA substrate with single-base-pair resolution (left panel) [41,91]. In this approach, a single DNA molecule containing a 1200 bp hairpin is attached to a super-paramagnetic bead at one end and to a coverslip surface at the opposite end. The position of the bead attached to the DNA is monitored in real-time by imaging it using a CCD camera. These images change (right middle panel) as the DNA molecule is affected by enzyme action and the images are correlated with changes in the Z-height when compared to the reference bead (R), which is fixed to the coverslip surface. During the assay, the position of the beads is carefully controlled by the application of a magnetic field (the magnetic tweezers), with the amount of force applied being proportional to the strength of the field. If an enzyme binds to the fork and catalyzes duplex unwinding, this is visualized as an increase in Z-height as a function of time and is interpreted as fork reversal (top right panel). In contrast, if an enzyme binds to the fork and catalyzes DNA annealing, this is observed as a decrease in Z-height as a function of time and this is interpreted as fork regression (bottom right panel). (**B**,**C**) In situ construction of DNA substrates relevant to fork transactions. (**B**) The construction of forks with gaps in the leading or lagging strands is facilitated by the application of force, which results in complete unzippering of the hairpin. This is followed by the introduction of oligonucleotides that anneal, in separate reactions, to the opposing fork arms. When the force is decreased, forks with gaps in either the nascent leading or lagging strand arms are revealed. (**C**) The assembly of a fork with 3 duplex arms. Here, a primer (red) is annealed to the ssDNA arm close to the coverslip surface (step 1). Then, as in panel (**B**), the hairpin is completely unzippered by the magnetic tweezers. This is followed by the introduction of a second primer (green), which anneals to a position several hundred base pairs distal to the first (step 2). When the force is decreased, the fork structure reforms with primers annealed to opposing arms (step 3). When DNA polymerase and dNTPs are introduced, the primers are extended, resulting in the formation of a fork with three 600 bp duplex arms (step 4).

**Figure 4 ijms-23-08613-f004:**
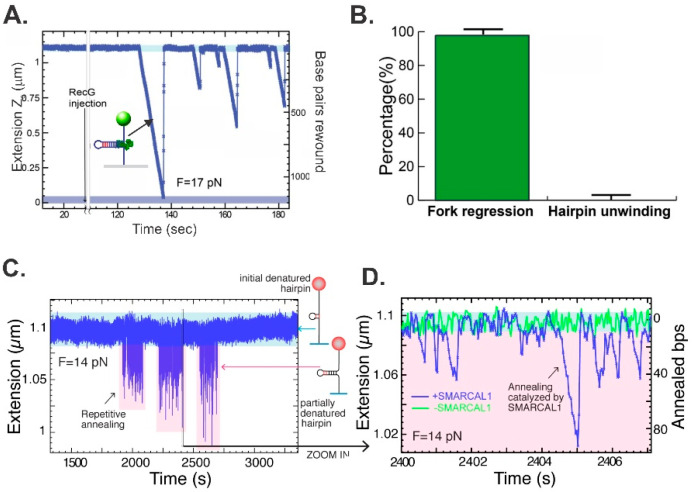
Kinetics and directionality of fork arm reannealing. (**A**,**B**) Reactions catalyzed by *E. coli* RecG [41]. (**C**,**D**) The reaction catalyzed by SMARCAL1 [46]. Representative time courses for the annealing reaction for each enzyme are shown in panels (**A**,**C**). The DNA substrate is the hairpin from Figure 3A. (**B**) RecG catalyzes fork regression only. Analysis of annealing reactions by RecG. Panels (**C**,**D**) are reprinted from *Cell Reports*, 3 (6), R. Betous, F. B. Couch, A. C. Mason, B. F. Eichman, M. Manosas and D. Cortez, *Substrate-Selective Repair and Restart of Replication Forks by DNA Translocases*, pages 1958–1969, Copyright (2013), with permission from Elsevier.

**Figure 5 ijms-23-08613-f005:**
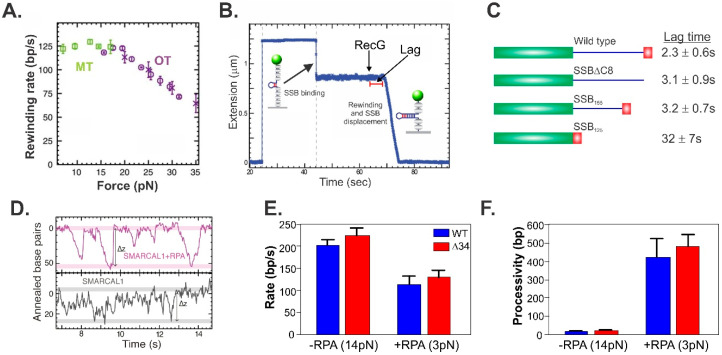
Single-strand DNA binding proteins affect fork regression enzymes differently. (**A**–**C**) RecG; (**D**–**F**) SMARCAL1. (**A**) RecG drives annealing against forces as high as 35 pN. (**B**) RecG couples SSB displacement to DNA annealing (rewinding). (**C**) The linker region of SSB is critical for efficient displacement by RecG. The mutants used are SSBΔC8 (last 8 residues deleted), SSB_155_ (22 residues deleted from the linker), and SSB_125_ (entire linker deleted). (**D**) RPA enhances the minimal processivity of SMARCAL1. (**E**,**F**) RPA enhances the processivity of SMARCAL1 and does not impact the annealing rate. SMARCAL1-Δ34 is a mutant that lacks the RPA binding domain. Panels (**D**–**F**) are adapted from *Cell Reports*, 3 (6), R. Betous, F. B. Couch, A. C. Mason, B. F. Eichman, M. Manosas and D. Cortez, *Substrate-Selective Repair and Restart of Replication Forks by DNA Translocases*, pages 1958–1969, Copyright (2013), with permission from Elsevier.

**Figure 6 ijms-23-08613-f006:**
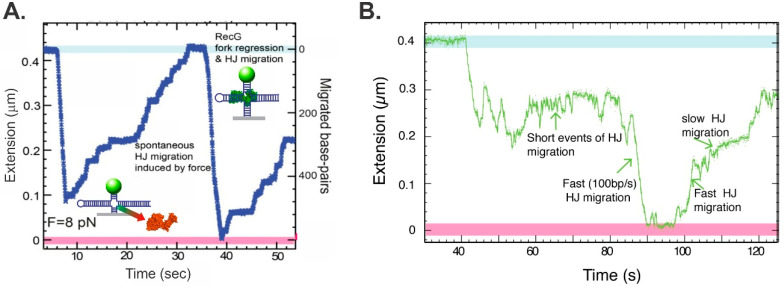
Extrusion of a Holliday junction, a key intermediate in fork regression. (**A**,**B**) reactions catalyzed by *E. coli* RecG [41] and SMARCAL1 [46], respectively. The DNA substrate for each enzyme was constructed as in Figure 3C. Panel (**B**) is reprinted from *Cell Reports*, 3 (6), R. Betous, F. B. Couch, A. C. Mason, B. F. Eichman, M. Manosas and D. Cortez, *Substrate-Selective Repair and Restart of Replication Forks by DNA Translocases*, pages 1958–1969, Copyright (2013), with permission from Elsevier.

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
