# Peer review of "The Biochemical Mechanism of Fork Regression in Prokaryotes and Eukaryotes—A Single Molecule Comparison"

_ijms, 2022, doi:10.3390/ijms23158613_

Round 1

Reviewer 1 Report

The review by Piero R. Blanco presents an appealing comparison of the bacterial helicase RecG and the eukaryotic SMARCAL1 in fork reversal at the single-molecule level. The manuscript reads well and is informative, with a clear description of the assay, and a side-by-side evaluation of published results.

I have just few suggestions:

If figures are coming from other sources, indicate that permission has been obtained for reproducing them here.

Page2 second paragraph: indicate that broken forks could result from DNA breaks.

Page5 third paragraph, 7th line: the sentence is as confusing as the nomenclature of SMARCAL1…, please clarify (the sentence)

Page5 third paragraph: clarify what is meant by ‘bulk-phase’

Page5 fourth paragraph, 4th line: last sentence is not clear, is there any typo?

Page5 fourth paragraph, 7th line: typo ‘doe’

Page5 fourth paragraph: useful in this paragraph to indicate what could be the consequences of the differences between RecG and SMARCAL1

Page 8 and page 9: use consistent units for the Kd.

Figure 6 legend: typo in the last point-brackets

Author Response

Response to reviewer 1

If figures are coming from other sources, indicate that permission has been obtained for reproducing them here.

Response: The permission has been obtained and is indicated in each figure.

Page2 second paragraph: indicate that broken forks could result from DNA breaks.

Response: This has been indicated.

Page5 third paragraph, 7th line: the sentence is as confusing as the nomenclature of SMARCAL1…, please clarify (the sentence)

Response: This has been altered.

Page5 third paragraph: clarify what is meant by ‘bulk-phase’

Response: No explanation is needed. Bulk-phase typically refers to reactions in a test tube with on average 108 molecules of enzyme present, which is different from one molecule.

Page5 fourth paragraph, 4th line: last sentence is not clear, is there any typo?

Response: There is no typo.

Page5 fourth paragraph, 7th line: typo ‘doe’

Response: This has been corrected as requested.

Page5 fourth paragraph: useful in this paragraph to indicate what could be the consequences of the differences between RecG and SMARCAL1

Response: This has been addressed and attributed to the OB-fold in RecG.

Page 8 and page 9: use consistent units for the Kd.

Response: This has been corrected as requested.

Figure 6 legend: typo in the last point-brackets

Response: This has been corrected as requested.

Reviewer 2 Report

The author compared the action of prokaryote RecG and eukaryote SMARCAL1 in the fork regression. It is always true that a purified enzyme may require additional factors and/or post-translational modifications to enhance its catalytic activity. But, in the review article, I hope the author discusses the biological meaning of mechanistic differences between RecG and SMARCAL1 and proposes constructive future directions in this research field.

Major points:

1.      It is better to provide something more constructive in the bottom line of the Abstract.

2.      P4. Can you discuss the possibility that there is a RecG ortholog in eukaryotes? If it is unlikely, can the author discuss why RecG is only present in prokaryotes?

3.      The terminologies: unwinding and rewinding, reversal and regression, and annealing are confusing.

4.      Figure 5C. The author should explain SSB∆C8, SSB155, and SSB125 data in the main text.

5.      Figure 5E and F. The author should explain ∆38 in the main text. It is impossible to define the effect of RPA from these graphs because not only RPA (+/–) but also the force (14 or 3pN) are different.

6.      I hope the author discusses the future directions in this research field in the last part of this review article.

7.      It is better to increase the resolution of figures.

Minor points:

1.      Figure 1D. It is better to explain what green lines represent. I cannot see the difference between “Chicken foot intermediate” and “Holliday junction”

2.      P3, L24. “inappropriately loaded nucleosomes” Can the author explain what they are?

3.      P3, L17 from the bottom. “three pathways of recombination” Can the authors briefly explain the three pathways?

4.      P5, L23 from the bottom. “as RecG doe as shown by footprinting” must be “as RecG do as shown by footprinting”

5.      P5, L19 from the bottom. Can the author explain in more detail that “RNA inhibits SMARCAL1 in a substrate-dependent manner”?

6.      Figure 3 legend. “, I n separate reactions,” must be “, in separate reactions,”

7.      P9, L21. “It s worth noting that” must be “It is worth noting that”

8.      P9, L23. “For wild type. The lag is ~2 sec” must be “For wild type, the lag is ~2 sec”

Author Response

Response to reviewer 2:

The author compared the action of prokaryote RecG and eukaryote SMARCAL1 in the fork regression. It is always true that a purified enzyme may require additional factors and/or post-translational modifications to enhance its catalytic activity. But, in the review article, I hope the author discusses the biological meaning of mechanistic differences between RecG and SMARCAL1 and proposes constructive future directions in this research field.

Major points:

  1. It is better to provide something more constructive in the bottom line of the Abstract.

Response: This has been changed to “Like many other eukaryotic enzymes, SMARCAL1 may require additional factors and/or post-translational modifications to enhance its catalytic activity whereas RecG can drive fork regression on its own.”

  1. Can you discuss the possibility that there is a RecG ortholog in eukaryotes? If it is unlikely, can the author discuss why RecG is only present in prokaryotes?

Response: This has been done as requested. There is a RecG ortholog in plants but it functions in mitochondria and plastids only.

  1. The terminologies: unwinding and rewinding, reversal and regression, and annealing are confusing.

Response: I agree. It has been made more confusing by other groups in the field as they use the terminology interchangeably. For clarity, unwind = strand separate; rewind = to anneal or reform the duplex. In the single molecule assay it is technically annealing one measures but the end point is a rewound duplex. Finally, I have defined in word and figure the directions of reversal and regression. The earliest definitions came from work by Ken Kruezer in the T4 field.

  1. Figure 5C. The author should explain SSB∆C8, SSB155, and SSB125 data in the main text.

Response: These definitions were already in the text but I added them to figure 5 legend as well.

  1. Figure 5E and F. The author should explain ∆38 in the main text. It is impossible to define the effect of RPA from these graphs because not only RPA (+/–) but also the force (14 or 3pN) are different.

Response: The reviewer is correct. I had great difficulty interpreting the original article and have done the best I can with the data available to me.

  1. I hope the author discusses the future directions in this research field in the last part of this review article.

Response: This has been addressed as requested.

  1. It is better to increase the resolution of figures.

Response: Higher resolution figures were obtained from Dr. Cortez.

Minor points:

  1. Figure 1D. It is better to explain what green lines represent. I cannot see the difference between “Chicken foot intermediate” and “Holliday junction”

Response: The explanation of different coloured lines is in the legend: “Black strands, parental DNA; blue and orange dashed strands, nascent leading, and lagging strands, respectively; green strands are nascent annealed parental strands.” A chicken foot and Hj are essentially the same thing, just different terminology used in the field.

  1. P3, L24. “inappropriately loaded nucleosomes” Can the author explain what they are?

Response: Nucleosomes and NAPs should be removed from the DNA ahead of advancing forks and reloaded once an intact fork has moved on. Inappropriately loaded would refer to loading before this has happened causing issues for the fork rescue process.

  1. P3, L17 from the bottom. “three pathways of recombination” Can the authors briefly explain the three pathways?

Response: These are the RecBCD; RecE and RecF pathways. This has been added to the text,

  1. P5, L23 from the bottom. “as RecG doe as shown by footprinting” must be “as RecG do as shown by footprinting”

Response: This has been corrected.

  1. P5, L19 from the bottom. Can the author explain in more detail that “RNA inhibits SMARCAL1 in a substrate-dependent manner”?

Response: I think the reviewer is referring to RPA “While RPA stimulates SMARCAL1 fork reversal activity when it is bound to an ssDNA gap on the leading template strand, it inhibits SMARCAL1 when bound to a replication fork with an ssDNA gap on the lagging strand (46). This difference may be related to the polarity of fork binding by RPA as this protein prefers forks with 3’-tails (89,90).”

The last sentence explains why.

  1. Figure 3 legend. “, I n separate reactions,” must be “, in separate reactions,”

Response: This has been corrected.

  1. P9, L21. “It s worth noting that” must be “It is worth noting that”

Response: This has been corrected.

  1. P9, L23. “For wild type. The lag is ~2 sec” must be “For wild type, the lag is ~2 sec”

Response: This has been corrected.

Round 2

Reviewer 2 Report

The author responded to my concerns.